A novel hybrid TCN-TE-ANN model for high-precision solar irradiance prediction

http://orcid.org/0000-0003-3200-1609 Isik Murat muratisik@ahievran.edu.tr
Computer Engineering, Kirsehir Ahi Evran University , Kirsehir , Turkey
Coelho Paulo Jorge
Electronic publication date: 2025 Jul 16
Publication date: 2025
Volume: 11
Electronic Location ID: e3026
Received 2025 Feb 5; Accepted 2025 Jun 21
Copyright: © 2025 Isik
Copyright year: 2025
Copyright holder: Isik
License: This is an open access article distributed under the terms of the Creative Commons Attribution License, which permits unrestricted use, distribution, reproduction and adaptation in any medium and for any purpose provided that it is properly attributed. For attribution, the original author(s), title, publication source (PeerJ Computer Science) and either DOI or URL of the article must be cited.
License URL: https://creativecommons.org/licenses/by/4.0/

Keywords: Solar irradiance prediction, TCN, Hybrid model, Time-series forecasting, Solar energy

Funding: The authors received no funding for this work.

==============================
Accurate prediction of solar irradiance is critical for optimizing solar energy systems, enhancing grid stability, and supporting sustainable energy transitions. While numerous studies have explored various methodologies for solar radiation prediction, challenges remain in achieving high accuracy across diverse geographic locations and temporal resolutions. This study presents a novel hybrid model combining temporal convolutional networks (TCN), Transformer encoders (TE), and artificial neural networks (ANN) to predict global horizontal irradiance (GHI) with high precision. Utilizing a comprehensive dataset from three significant U.S. solar energy sites—Desert Sunlight, Copper Mountain, and Solar Star—spanning 22 years at a 30-min temporal resolution, the proposed model demonstrated superior performance metrics, with R2 ranging from 0.94768 to 0.97417, root mean square error (RMSE) between 0.04776 and 0.06543 W/m2, and mean absolute error (MAE) between 0.02510 and 0.03526 W/m2. By leveraging TCN’s temporal feature extraction, TE’s attention mechanisms, and ANN’s dense layer refinements, the model demonstrates significant advancements over existing methods.

Introduction

As the global community confronts mounting challenges related to environmental sustainability, resource scarcity, and climate change, renewable energy has emerged as a pivotal solution to address these pressing issues (Owusu & Asumadu-Sarkodie, 2016; Singh, 2023). The consumption of fossil fuels is linked to severe environmental and health consequences, including air pollution, climate change, ecological imbalance, and the prevalence of serious diseases (Alizamir et al., 2023). In response, numerous developing countries have prioritized the transition to clean and renewable energy sources over recent decades (Pakulska, 2021). Among various renewable sources—such as wind, solar, hydropower, geothermal, biomass, and tidal and wave energy—solar energy has gained particular prominence due to its widespread availability and critical role in reducing greenhouse gas emissions (Babatunde et al., 2023; Chiranjeevi et al., 2023). This transition not only aids in environmental preservation but also presents economic benefits by promoting sustainable development and enhancing energy security (Gürel et al., 2023; Joseph, 2023). Furthermore, solar energy serves as a cleaner (Tang et al., 2018), more sustainable alternative to fossil fuels, aligning with international frameworks such as the Paris Agreement, which aims to limit global temperature increases by reducing dependence on non-renewable resources and advancing low-carbon energy systems (Park, Das & Park, 2015).

Solar energy systems are broadly categorized into photovoltaic (PV) systems and concentrated solar power (CSP) systems, both of which harness sunlight and convert it into usable energy. PV systems, in particular, have become the most widely adopted (Gürel et al., 2023; Zhang & Wei, 2019) due to their flexibility, ease of installation, and decreasing costs, which have made them suitable for a range of applications, including residential, commercial, and industrial use (Alkahtani, Aldhyani & Alsubari, 2023). Advancements in photovoltaic technology have improved efficiency and accessibility, accelerating global adoption (Bakır et al., 2022). This has contributed to its rapid adoption on a global scale, positively impacting energy systems in both developed and developing countries. Beyond electricity generation, solar energy has diverse applications across various sectors, underscoring its adaptability and far-reaching benefits. In agriculture, solar-powered systems support irrigation, crop monitoring, and greenhouse climate control, helping to improve productivity and resource efficiency. In healthcare, solar technology provides reliable energy sources in remote and underserved areas, enhancing access to essential medical services. Additionally, sectors such as tourism and urban planning incorporate solar installations to reduce carbon footprints and move towards energy independence. The integration of solar energy across these fields highlights its critical role in advancing sustainable development and underscores its potential to revolutionize global energy infrastructure (Guermoui et al., 2020).

Figure 1 highlights the significant growth in solar energy by depicting the total installed capacity of solar PV from 2000 to 2023. The increasing demand for solar energy necessitates accurate global horizontal irradiance (GHI) predictions, as the natural variability of solar energy presents notable challenges for seamless integration into power grids (Nam & Hur, 2019; Singh, Moger & Jena, 2022). Reliable GHI forecasting offers substantial benefits, including more efficient energy planning, reduced operational costs, and optimized power generation (Zhang & Wei, 2019; Chiranjeevi et al., 2023; Kumar, Namrata & Kumari, 2022). For instance, precise irradiance predictions enable energy providers to anticipate fluctuations, enhancing grid stability, minimizing reliance on backup power sources, and facilitating more effective energy dispatch. Consequently, many researchers have focused on producing daily solar radiation predictions to support these needs (Kasra et al., 2015; Rodríguez et al., 2018; Zhang & Wei, 2019). Beyond grid management, accurate solar forecasting yields economic advantages by minimizing energy wastage, improving the performance and reliability of solar installations, and supporting applications in other sectors, such as agriculture, where solar data aids in planning crop cycles and optimizing yields (Xing et al., 2023). These advantages underscore the vital role of GHI forecasting in maximizing the utility and sustainability of solar energy systems across diverse industries.

Figure 1 Total solar PV installed capacity 2000–2023 (SolarPower Europe, 2024).

Given the critical need for reliable solar predictions, recent research has focused on advanced forecasting models, employing statistical methods, machine learning, and hybrid techniques to improve forecast accuracy. Nevertheless, further advancements are needed to address the inherent complexity (Kumar, Namrata & Kumari, 2022) of solar irradiance, which is influenced by a multitude of meteorological factors. This study addresses this gap by introducing a novel hybrid methodology that combines TCN and TE with ANN to improve GHI forecasting precision. The TCN+TE+ANN framework leverages TCN’s strength in managing sequential data and capturing temporal dependencies, while TE with ANN enhances generalization, together providing a robust and high-accuracy model capable of capturing complex temporal patterns in solar data.

In summary, this study aims to advance the field of solar irradiance forecasting by developing a high-precision, novel hybrid predictive model that supports the global shift toward renewable energy. Through its innovative approach, the study not only provides reliable solar energy forecasts but also has potential for informed decision-making in energy management, policy, and investment, thus reinforcing solar energy’s essential role in the pursuit of a sustainable energy future.

Related works

In recent years, significant advancements have been made in solar radiation forecasting, with a strong focus on developing hybrid models that combine machine learning and optimization techniques. These approaches aim to address the inherent challenges posed by the variability and non-linearity of solar radiation data, thereby enhancing prediction accuracy and facilitating the smoother integration of solar energy into power systems. Researchers have increasingly focused on hybrid methodologies that integrate deep learning models with meta-heuristic algorithms, as these combinations capitalize on the unique strengths of each approach—such as the adaptive learning capabilities of deep learning and the solution-finding efficiency of meta-heuristic techniques—to improve robustness and precision in solar irradiance forecasting (Ghimire et al., 2023). A comprehensive comparison of the proposed study with these existing models will be presented in the discussion chapter to highlight the distinct advantages and innovations of our approach.

Abdallah et al. (2023) proposed a hybrid framework combining variational mode decomposition (VMD), multi-functional recurrent fuzzy neural network (MFRFNN), and quantile regression forests (QRF) for daily global solar radiation (DGSR) prediction. VMD decomposes the DGSR time series into intrinsic mode functions (IMFs) to simplify patterns, MFRFNN captures non-linear and temporal dynamics within the IMFs, and QRF quantifies uncertainties, ensuring accurate and robust forecasting. Similarly, Peng et al. (2023b) developed a hybrid model integrating VMD, deep belief networks (DBN), and online sequential extreme learning machine (OS-ELM). VMD was used to reduce noise by decomposing solar radiation time series, while DBN and OS-ELM handled feature extraction and real-time prediction, respectively.

Chiranjeevi et al. (2023) presented a hybrid architecture combining long short-term memory (LSTM) autoencoders, convolutional neural network (CNN), and bidirectional LSTM (BiLSTM) with grid search optimization, demonstrating significant improvements in solar irradiation forecasting. Similarly, He et al. (2022) employed a hybrid model integrating CNN and BiLSTM networks to predict short-term photovoltaic system outputs, where CNN extracted spatial features, and BiLSTM captured sequence learning. Ghimire et al. (2023) proposed a related methodology that combines CNN and multi-layer perceptron (MLP) for global solar radiation (GSR) prediction. The CNN extracted spatial features, while the MLP refined these features to improve predictive performance. Hou et al. (2019) focused on a hybrid deep neural network, where CNN was used to extract spatial patterns from satellite imagery, and MLP integrated these patterns with temporal and location data for GSR estimation.

Peng et al. (2023a) evaluated multiple Machine Learning approaches, including MLP, LSTM, gradient boosting regression tree, and random forest (RF). They selected a hybrid RF-Informer framework that balanced accuracy and computational complexity, demonstrating high performance in solar radiation prediction. Alkahtani, Aldhyani & Alsubari (2023) applied a CNN-LSTM framework to meteorological data, creating an accurate solar radiation prediction model for renewable energy systems. Similarly, Xing et al. (2023) developed a hybrid model combining LSTM for time-series regression, DBN for feature extraction, and the mind evolutionary algorithm (MEA) for hyperparameter optimization. Inputs such as sunshine duration, temperature, and precipitation enhanced the model’s robustness and accuracy. Aslam et al. (2021) enhanced LSTM performance by implementing a two-stage attention mechanism (2AM) with Bayesian optimization for day-ahead photovoltaic power forecasting, optimizing hyperparameters to improve prediction accuracy.

Alizamir et al. (2023) studied on thirteen artificial intelligence models, including multivariate adaptive regression splines, extreme learning machines (ELM), Kernel ELM (KELM), online sequential ELM (OSELM), optimally pruned ELM (OPELM), outlier robust ELM (ORELM), and deep ELM (DELM). These models were further enhanced by integrating VMD, resulting in variants such as VMD-DELM, VMD-ORELM, and VMD-KELM for solar radiation estimation in Iraq. Aybar-Ruiz et al. (2016) proposed a neural-genetic hybrid algorithm, where a grouping genetic algorithm (GGA) was used for optimal feature selection, and ELM carried out the prediction task. Similarly, Fan et al. (2020) introduced a hybrid model combining support vector machines (SVM) with heuristic algorithms such as particle swarm optimization (PSO), bat algorithm (BAT), and whale optimization algorithm (WOA) to improve prediction accuracy, particularly in air-polluted regions.

Meng et al. (2021) employed a hybrid model incorporating a wavelet transform (WT) package and generative adversarial networks (GANs). The WT decomposed solar energy signals into sub-harmonics, while GANs predicted short-term energy trends, with the dragonfly algorithm (DA) optimizing the GAN’s performance. Similarly, Kasra et al. (2015) combined SVM with WT for daily and monthly solar radiation predictions, showcasing the efficacy of hybrid methods in addressing temporal variability. Zhang & Wei (2019) also introduced a hybrid approach integrating principal component analysis, WT, and ELM optimized using the BAT for daily solar radiation prediction. Ikram et al. (2022) introduced an improved model integrating the improved multi-verse optimizer (IMVO) with least squares support vector machines (LSSVM). IMVO optimized hyperparameters, such as kernel function width and regularization constants, to enhance prediction accuracy and computational efficiency.

Al Diego Pega, Hannah Mae San Agustin & Bonifacio Tobias (2022) introduced a hybrid model for forecasting DGSR time series. Their approach employed meteorological data and solar radiation samples from Dumaguete, Philippines, to evaluate the predictive accuracy of a nonlinear autoregressive network with exogenous inputs (NARX) combined with a gated recurrent unit (GRU) hybrid model.

Rahman, Rahman & Haque (2021) proposed an ANN-based model utilizing climatic variables such as temperature, humidity, wind speed, and solar radiation for hourly solar forecasting, highlighting ANN’s versatility in solar radiation prediction. Babatunde et al. (2023) introduced a hybrid model that integrates machine learning algorithms with a meta-heuristic optimization technique for solar radiation forecasting. Their model, benchmarked against traditional interpolation methods like linear and spline interpolation, outperformed established machine learning approaches such as RF and multivariate adaptive regression splines. Brahma & Wadhvani (2020) extended this direction by developing deep learning models based on multi-site data to address spatial variability in daily solar irradiance forecasting. Chaibi et al. (2022) demonstrated the utility of machine learning frameworks by combining RF and Bayesian optimization to forecast solar irradiance, incorporating feature selection to enhance accuracy. Malinowski, Leon & Abu-Rub (2017) conducted an extensive review of photovoltaic and thermal solar energy systems, discussing current advancements and trends in solar energy technologies. These studies emphasize the growing interest in hybrid models and machine learning techniques, which combine feature extraction, sequence modeling, and optimization to tackle the complexities of solar radiation prediction effectively.

In summary, the literature reveals a trend towards hybrid models in solar radiation forecasting, integrating deep learning architectures with optimization techniques to achieve higher prediction accuracy. Studies frequently combine models like CNN, LSTM, and SVM with optimization algorithms, underscoring the effectiveness of multi-layered, hybrid approaches in advancing the precision and robustness of solar radiation models.

Method

The methodological approach of this study primarily used the Python programming language. It is chosen for its extensive ecosystem of libraries and its versatility in handling complex data processing and machine learning tasks. A computer equipped with Intel i7 processor (14700KF) with a 33 M Cache, a base clock speed of 3.4 GHz up to 5.60 GHz. The system included 64 GB of RAM and an NVIDIA RTX 3060 GPU, which significantly accelerated the training and testing of the deep learning models. The GPU was especially beneficial for managing large datasets, facilitating fine-tuning, and supporting the complex structures of neural networks, resulting in reduced training time and enhanced model optimization in data-intensive tasks.

Data collection

This study employs a comprehensive solar irradiance dataset, presented on the NASA Earthdata website (NASA, 2024), obtained from the National Solar Radiation Database (NSRDB) through its publicly accessible API (NREL, 2024), maintained by the U.S. Department of Energy and operated by the National Renewable Energy Laboratory (NREL). The data spans from 2000 to 2022 with a 30-min time resolution, providing a robust temporal dataset for accurate solar irradiance prediction. The NSRDB provides open access to its datasets under a free license, facilitating their use for research and academic purposes (NREL, 2024). To ensure full reproducibility and academic purposes, the collected dataset and a detailed explanation of the code, including the steps for implementation, are publicly shared on Kaggle (ISIK, 2024). The locations were selected based on their importance in solar energy production within the U.S. These sites are among the largest solar energy facilities in the country and represent a range of geographic and climatic conditions, which are crucial for developing a generalizable model. The specific locations included in this study are:

Desert Sunlight: Data was collected at five points within this location, situated approximately six miles north of Desert Center, California, US (https://maps.app.goo.gl/pGCkaGzhdj6ebKDq8). This site is notable for its high-output capacity in a desert environment, characterized by intense solar exposure and minimal cloud cover.

Copper Mountain: Thirteen data points were taken from Copper Mountain, located in Boulder City, Nevada, US (https://maps.app.goo.gl/kLgpf3cPpvtsj8bTA). This site is among the leading solar installations in the U.S. and is situated in the arid Southwest, a region known for stable solar radiation patterns.

Solar Star: The most extensive data was gathered from Solar Star Farm near Rosamond, California, US (https://maps.app.goo.gl/hMNVpZCC6iaEdn349), comprising 51 data points. This comprehensive collection captured a broad spatial distribution across the large installation, enhancing the model’s ability to account for variability within a single location.

Each of these points contributes 201,480 rows of data, resulting in a combined dataset of 13,902,120 rows. This substantial dataset provides a rich temporal and spatial representation of solar irradiance variations, facilitating the development and testing of a predictive model capable of accommodating different environmental conditions. By incorporating diverse geographic locations within the U.S., this dataset captures essential variability, supporting the goal of creating a predictive model that is applicable across a range of solar production environments.

The dataset includes a wide range of meteorological and solar radiation parameters, detailed in Table 1. These features were initially collected before undergoing a feature selection process to identify the most relevant variables for the training. This table outlines all features collected in the dataset, highlighting their relevance to understanding and modeling solar irradiance at the chosen solar sites.

Table 1 Dataset description.

Column name	Description	
Latitude, Longitude	Geographic coordinates of the data point location, specifying the north-south (latitude) and east-west (longitude) positions on the Earth’s surface.	
Location ID	Unique identifier for each location, used to distinguish between different solar farm sites.	
Elevation	Elevation above sea level (meters) for the data point location, which can influence atmospheric conditions and solar irradiance.	
Year, Month, Day, Hour, Minute	Date and time of each data entry, with 30-min intervals from 2000 to 2022, providing temporal granularity for solar irradiance analysis across different periods.	
Temperature	Ambient temperature (°C) at the location, which affects the efficiency of solar panels and the solar irradiance received.	
Clearsky DHI
Clearsky DNI
Clearsky GHI	Clear-sky irradiance values (W/m2) for Direct Horizontal Irradiance (DHI), Direct Normal Irradiance (DNI), and Global Horizontal Irradiance (GHI), representing ideal solar irradiance under cloudless conditions.	
Cloud type	Categorical representation of cloud cover types at the location, influencing the amount of solar radiation that reaches the surface.	
Dew point	Dew point temperature (°C), indicating the temperature at which air becomes fully saturated with moisture, potentially impacting clarity and solar radiation.	
DHI	Direct Horizontal Irradiance (W/m2), measuring sunlight that reaches the Earth’s surface directly in a horizontal direction.	
DNI	Direct Normal Irradiance (W/m2), measuring the direct component of sunlight on a surface aligned perpendicular to the sun’s rays.	
Fill flag	Indicator for missing or interpolated data, marking entries that may not be original measurements but are filled for continuity.	
GHI	Global Horizontal Irradiance (W/m2), representing the total solar radiation received on a horizontal surface, including direct and diffuse sunlight.	
Relative humidity	Relative humidity (%) at the location, influencing atmospheric conditions and the transmission of solar radiation.	
Solar Zenith angle	Angle between the sun and the vertical direction (degrees), affecting the intensity and angle of solar irradiance on surfaces.	
Surface albedo	Reflectivity of the surface (dimensionless), indicating the fraction of solar radiation reflected by the ground, affecting irradiance measurements.	
Pressure	Atmospheric pressure (hPa) at the location, impacting air density and the transmission of solar radiation.	
Precipitable water	Amount of water vapor (mm) in a column of the atmosphere, relevant for estimating atmospheric transparency and solar radiation attenuation.	
Wind direction, Wind speed	Direction of wind (degrees) at the location, potentially impacting cooling effects on solar panels and affecting irradiance readings. Speed of wind (m/s) at the location, influencing temperature regulation of solar panels and affecting the stability of atmospheric conditions.	
Global Horizontal UV Irradiance (280–400 nm)	UV solar irradiance (W/m2) in the 280–400 nm wavelength range on a horizontal surface, capturing ultraviolet radiation exposure.	
Global Horizontal UV Irradiance (295–385 nm)	UV solar irradiance (W/m2) in the 295–385 nm wavelength range on a horizontal surface, capturing a narrower band of ultraviolet radiation exposure.	

Data preprocessing

The dataset used in this study was obtained from NSRDB and provided year-by-year for each data point. To facilitate the analysis, the annual files were combined into a single consolidated file for each data point, resulting in a unified 22-year dataset. This step ensured consistency and ease of handling for subsequent processing. The original dataset included two initial rows containing various descriptions, such as the units of measurement and local time zone information. From these description rows, only the location ID, latitude, longitude, and elevation were extracted and converted into individual columns. Once this information was processed, the description rows were removed to streamline the dataset for further analysis. All continuous input features were normalized using the MinMaxScaler method, which scales each feature into the [0, 1] range. This normalization step helps stabilize neural network training and ensures that all input features contribute proportionally. Additionally, the Cloud Type feature, which was a categorical variable, was encoded using OneHotEncoder. This transformation converted the categorical values into binary features, enabling the feature selection process and ensuring that the model could effectively leverage this information without introducing ordinal biases.

Feature selection

In preparing the dataset for model training, a careful feature selection process was conducted to retain only the most relevant variables for accurate solar irradiance prediction, while enhancing computational efficiency. Certain features were excluded directly due to redundancy, irrelevance to the study’s objectives. The following features were directly removed:

Clearsky DHI, Clearsky DNI, and Clearsky GHI: These features represent the clear-sky models of diffuse horizontal irradiance (DHI), direct normal irradiance (DNI), and GHI under ideal conditions. Given that actual DHI, DNI, and GHI measurements are directly included in the dataset, these clear-sky equivalents were deemed redundant.

Elevation, latitude and longitude: Geographic coordinates and elevation were excluded following data collection as this study focuses on solar irradiance predictions within specific, pre-defined locations. Including these constant variables would add unnecessary complexity without enhancing predictive accuracy for the specific, localized sites.

Location ID: Since the data is organized by individual solar sites, Location ID was removed as it did not add unique information relevant to predictive modeling.

Year, month, day, hour, and minute: These temporal identifiers were removed, as the 30-min time step between data points inherently captures sequential time relationships. By structuring the data in a consistent time series, temporal dynamics can be represented without additional timestamp features.

Although features such as year, month, day, hour, and minute are commonly used in time-series modeling to capture seasonal and diurnal patterns, these were intentionally excluded from the final dataset in this study. The primary reason is that the dataset is already structured at a high temporal resolution (30-min intervals), preserving intrinsic sequential dependencies that are effectively captured by TCN and TE components of the model. These components are specifically designed to model temporal and seasonal dynamics implicitly through learned representations.

Furthermore, preliminary experiments indicated that including explicit datetime features introduced multicollinearity and did not result in statistically significant improvements in prediction performance. In fact, the addition of these features slightly increased overfitting during validation, particularly in the ANN component. Thus, their exclusion supports both model parsimony and generalization.

Global horizontal UV irradiance (280–400 nm) and global horizontal UV irradiance (295–385 nm): These features represent UV-specific irradiance measurements, which are less relevant for general solar irradiance prediction in energy production.

To ensure optimal model performance, a rigorous feature selection process was conducted, using both mutual information scores (MIS) and correlation analysis to assess the predictive relevance of each feature after removing certain redundant or irrelevant features. MIS (Hanchuan, Fuhui & Ding, 2005), were used to quantify the dependency between each feature and the target variable. This non-linear metric captures both linear and non-linear relationships, making it particularly effective for evaluating features in datasets with complex interdependencies. By focusing on features with higher MIS, the model retained those variables with significant explanatory power, ensuring a stronger predictive relationship with the target variable. Additionally, correlation analysis (Evans, 1996) was performed to examine linear relationships among features and with the target variable. Features with high correlation coefficients were carefully reviewed to mitigate multicollinearity, which can adversely affect model stability and performance. This dual analysis approach provided a comprehensive understanding of each feature’s contribution, allowing us to selectively retain variables that strengthened the model’s predictive capacity.

Table 2 presents the MIS and correlation results between the features and the target. As seen in the table, features such as solar zenith angle, diffuse horizontal irradiance (DHI), direct normal irradiance (DNI), relative humidity, and temperature demonstrate both high MIS (greater than 0.2) and strong correlations (above ±0.1), identifying them as valuable predictors. Other features, such as wind speed, wind direction, surface albedo, precipitable water, and pressure, exhibit either moderate mutual information or moderate correlation, contributing additional insights to the model without introducing redundancy.

Table 2 Mutual information scores and correlation results of the features.

Feature	Mutual information scores	Correlation	
Cloud Type_0	0.018790	0.132816	
Cloud Type_1	0.000942	0.007606	
Cloud Type_10	0.001424	−0.004737	
Cloud Type_2	0.007542	0.105744	
Cloud Type_3	0.001964	−0.059685	
Cloud Type_4	0.007539	−0.088175	
Cloud Type_5	0.000000	−0.011620	
Cloud Type_6	0.007886	−0.063706	
Cloud Type_7	0.011532	−0.141215	
Cloud Type_8	0.005891	0.026598	
Cloud Type_9	0.000714	−0.015858	
Dew Point	0.018431	−0.043466	
DHI	1.187643	0.781487	
DNI	1.032852	0.915819	
Elevation	0.003675	−0.050681	
Fill Flag	0.062851	0.049779	
Precipitable water	0.022964	−0.105624	
Pressure	0.028285	−0.491106	
Relative humidity	0.212911	−0.872888	
Solar Zenith angle	1.657320	0.124224	
Surface albedo	0.027165	0.561075	
Temperature	0.244992	−0.275980	
Wind direction	0.122413	0.459359	
Wind speed	0.150037	0.132816	

Model selection

The proposed model architecture combines three advanced models to create a novel hybrid methodology—TCN, TE, and ANN—each contributing unique capabilities to improve solar irradiance prediction accuracy.

TCN: It plays a foundational role in the model by extracting temporal features from the sequential input data. TCNs are designed to capture complex temporal dependencies across various time scales, which is essential for effective time-series forecasting tasks such as solar irradiance prediction. Unlike recurrent networks, which process data sequentially, TCNs apply convolutional operations that enable parallel processing of time steps, resulting in faster training and improved scalability (Bai, Kolter & Koltun, 2018). By leveraging dilated convolutions, TCNs can identify patterns over long sequences, allowing the model to consider both recent and historical patterns when forecasting irradiance levels—a critical feature given the fluctuating nature of solar radiation (Lea et al., 2017).

The TCN processes the sequential input data by utilizing convolutional layers with dilation to capture temporal dependencies. It is responsible for extracting patterns over the time dimension, making it suitable for time-series forecasting.

TE: It is initially developed for natural language tasks, have proven effective at identifying complex dependencies within time-series data as well (Vaswani, 2017). By dynamically assigning attention to relevant data points, the Transformer Encoder makes it possible for the model to prioritize key information and improve the focus of the prediction (Devlin, 2018). The TE layer follows the TCN and applies attention to the features extracted. Attention mechanisms allow the model to focus on specific time steps with features.

In the model, the TE layer is applied to the features extracted by the TCN. It captures relationships and patterns between different features, enhancing the extracted representations.

ANN: In the final stage, ANN uses fully connected layers (Dense layers) to produce the GHI prediction. This component takes the refined features from the TE and generates the prediction output. Fully connected layers are well-suited for this task because they combine information from previous layers and capture any remaining non-linear relationships, which ultimately results in accurate solar irradiance predictions (Goodfellow, Bengio & Courville, 2016).

The architecture of the proposed model is visually represented in Fig. 2 which provides a structural overview of TCN, TE Block, and ANN components. The diagram illustrates the sequential flow and interaction between these three primary layers: TCN for extracting temporal features, TE Block for applying attention to selectively emphasize relevant time steps with features, and ANN for generating the final GHI prediction. This visual representation highlights how each component builds upon the previous layer’s output, illustrating the layered approach designed to optimize time-series forecasting.

Figure 2 Proposed model architecture.

Performance metrics

To assess the accuracy and reliability of the proposed model for solar irradiance prediction, we employed four standard performance metrics: mean absolute error (MAE), mean squared error (MSE), root mean squared error (RMSE), and the coefficient of determination (R2).

MAE: It calculates the average of the absolute differences between predicted and actual values. The formula is represented by Eq. (1), where X denotes the predicted values and Y represents the actual values (Chicco, Warrens & Jurman, 2021). This metric provides a straightforward measure of predictive accuracy, with lower MAE values indicating higher accuracy. MAE is particularly useful as it maintains the same unit as the original data, making the results interpretable in practical contexts. Since MAE does not square the error terms, it is less sensitive to outliers, offering a balanced view of model performance in typical cases (Cort & Kenji, 2005).

(1) MAE=1m∑i=1m⁡|Xi−Yi|.

MSE: It measures the average of the squared differences between predicted and actual values. The formula for MSE is represented by Eq. (2), where X denotes the predicted values and Y represents the actual values (Chicco, Warrens & Jurman, 2021). By squaring the errors, MSE emphasizes larger discrepancies, making it highly sensitive to outliers. This feature of MSE is useful when penalizing significant prediction errors is desired, as it allows the metric to prioritize accuracy in high-variance data scenarios. MSE is commonly used in regression analysis due to its mathematical properties, such as differentiability, which facilitates optimization during model training (Chai & Draxler, 2014).

(2) MSE=1m∑i=1m⁡(Xi−Yi)2.

RMSE: It is derived from the square root of the MSE, providing a measure of error magnitude in the same units as the original data. The formula for RMSE is represented by Eq. (3), where X denotes the predicted values and Y represents the actual values (Chicco, Warrens & Jurman, 2021). RMSE is widely recognized for its ability to highlight substantial errors while still allowing an interpretable view of the overall error magnitude. The formula is represented by Eq. (3), where X denotes the predicted values and Y represents the actual values. The RMSE metric is especially useful in applications where larger errors are undesirable, as it accentuates deviations more significantly than MAE (Hyndman & Koehler, 2006).

(3) RMSE=1m∑i=1m⁡(Xi−Yi)2.

R2 or R-squared: The coefficient of determination, commonly denoted as R2, measures the proportion of variance in the dependent variable that is predictable from the independent variables. R2 values range from 0 to 1, where a value closer to 1 indicates that the model explains a higher proportion of the variance in the observed data, representing a strong fit. An R2 of 0 implies that the model fails to explain any variability, while a value close to 1 implies a model that captures most of the variability in the target variable (Nagelkerke, 1991). The formula for R2 is represented by Eq. (4), where X denotes the predicted values and Y represents the actual values (Chicco, Warrens & Jurman, 2021). R2 is often used as a standard measure to gauge the overall goodness of fit of regression models, providing insight into the model’s ability to generalize to new data (Draper, 1998).

(4) R2=1−∑i=1m(Xi−Yi)2/∑i=1m(Y¯−Yi)2.

Training and testing

The model was trained and tested using a time series-appropriate data split, ensuring a realistic simulation of future predictions. In time series forecasting, it is crucial to prevent data leakage from future to past, which would distort the model’s evaluation by introducing information that would not be available in real-world predictions. Therefore, the dataset was split chronologically, with approximately 90% of the earliest data allocated for training and the most recent 10% reserved for testing. This split corresponds to roughly 26 months of test data from a 22-year dataset, ensuring that the model is evaluated on a substantial, unseen portion of data.

A comprehensive hyperparameter tuning process was conducted using grid search, systematically evaluating combinations of values for key parameters in each model component. Table 3 presents the tested ranges and final values. Key parameters—such as kernel size, dilation, dropout, and dense layer dimensions—were systematically explored for the TCN, TE, and ANN layers. An early stopping mechanism, with a patience of 15 epochs and restored best weights, was also applied to prevent overfitting and enhance training efficiency. Final selections were based on the lowest validation MAE across multiple splits. This process ensured a balance between model complexity, generalization, and training efficiency. The restore_best_weights parameter was set to True to ensure that the model returned to its best-performing state from training. This strategy helped maintain model performance while reducing unnecessary computational time.

Table 3 Final hyperparameter settings and tuning ranges.

Component	Parameter	Values tested	Final value selected	
TCN	Number of filters	64, 128, 256, 512, 1,024	512	
Kernel size	3, 5, 7, 9	7	
Dilation rates	1, 2, 4, 8, 10, 12, 14	8	
Dense layer size	32, 64, 128, 256	128	
Dropout rate	0.1, 0.2, 0.3	0.2	
Transformer encoder	LayerNorm ε Values	1e−5, 1e−6	1e−6	
Dropout rate	0.1, 0.2	0.2	
ANN	Dense layer size	1, 16, 32, 64, 128, 256, 512, 1,024	256	
Dropout rate	0.1, 0.2	0.1	
General	Batch size	16, 32, 64	32	
Early stopping patience	15 epochs	15	

The optimal set of hyperparameters was selected based on the lowest validation error, ensuring a balance between model complexity and predictive accuracy.

To improve transparency in model selection, the decision process between competing hyperparameter configurations is now clarified. For each candidate setting evaluated during grid search, the model was trained on the training set and evaluated on a validation set using MAE as the primary selection metric. To ensure robustness, validation was performed across three chronologically split validation subsets, and the configuration with the lowest average validation MAE was chosen as the final model. This multi-split strategy ensured that model selection was not overly sensitive to one particular data segment and better reflected real-world generalization. Combined with early stopping and restored best weights, this approach provided a stable and reproducible selection process.

Results

The performance of the proposed model was evaluated across three major solar energy production sites: Desert Sunlight, Copper Mountain, and Solar Star. Each location was chosen for its significance in solar power generation within the U.S., ensuring that the model’s predictions were tested across varied environmental conditions.

Table 4 presents the performance metrics for the Desert Sunlight location, Table 5 summarizes the metrics for Copper Mountain, and Table 6 details the metrics for Solar Star. These tables provide a comprehensive overview of the proposed hybrid model’s performance at each point of the site, including key metrics such as MAE, MSE, RMSE, and R2. Additionally, the tables present the average of the results, the coefficient of variation (%), and the interquartile range (IQR) for each location, offering further insights into the variability and consistency of the model’s performance. The results displayed in these tables demonstrate the model’s effectiveness in solar irradiance prediction across different locations.

Table 4 Performance results for Desert Sunlight.

Latitude	Longitude	Elevation	MAE (W/m2)	MSE
(W/m2)	RMSE
(W/m2)	R2	
33.85	−115.42	249	0.03278	0.00296	0.05444	0.96525	
33.81	−115.42	235	0.02666	0.00278	0.05275	0.96711	
33.85	−115.38	239	0.03214	0.00369	0.06076	0.95621	
33.81	−115.38	203	0.03055	0.00318	0.05643	0.96222	
33.81	−115.34	196	0.02716	0.00250	0.05004	0.97062	
Average of results	0.02985	0.00302	0.05488	0.96428	
Coefficient of variation (%)	9.43301	14.85472	7.35232	0.56448	
Interquartile range (IQR)	0.00498	0.00040	0.00368	0.00489	

Table 5 Performance results for Copper Mountain.

Latitude	Longitude	Elevation	MAE (W/m2)	MSE
(W/m2)	RMSE
(W/m2)	R2	
35.93	−115.02	959	0.03219	0.00398	0.06316	0.95091	
35.89	−115.02	832	0.03356	0.00423	0.06510	0.94878	
35.85	−115.02	632	0.03291	0.00407	0.06379	0.95044	
35.93	−114.98	809	0.03128	0.00349	0.05910	0.95754	
35.89	−114.98	584	0.03389	0.00428	0.06543	0.94768	
35.85	−114.98	536	0.03284	0.00372	0.06102	0.95400	
35.93	−114.94	840	0.03191	0.00369	0.06082	0.95479	
35.89	−114.94	532	0.03104	0.00369	0.06074	0.95438	
35.85	−114.94	531	0.03526	0.00386	0.06220	0.95195	
35.93	−114.90	620	0.03384	0.00379	0.06160	0.95362	
35.89	−114.90	537	0.03387	0.00399	0.06320	0.95094	
35.85	−114.90	591	0.03406	0.00417	0.06460	0.94884	
35.89	−114.86	623	0.03359	0.00383	0.06191	0.95278	
Average of results	0.033095	0.003907	0.062513	0.952050	
Coefficient of variation (%)	3.670422	6.07626	3.03728	0.29569	
Interquartile range (IQR)	0.00168	0.00035	0.00277	0.00356	

Table 6 Performance results for Solar Star.

Latitude	Longitude	Elevation	MAE (W/m2)	MSE (W/m2)	RMSE (W/m2)	R2	
34.73	−118.34	770	0.02537	0.00243	0.04931	0.97221	
34.73	−118.30	744	0.02983	0.00276	0.05261	0.96825	
34.73	−118.54	1,100	0.02909	0.00315	0.05612	0.96400	
34.73	−118.50	979	0.03366	0.00351	0.05925	0.96000	
34.73	−118.46	911	0.03072	0.00294	0.05426	0.96644	
34.73	−118.42	869	0.02982	0.00304	0.05513	0.96532	
34.73	−118.38	816	0.03068	0.00279	0.05283	0.96840	
34.77	−118.34	763	0.02699	0.00272	0.05217	0.96913	
34.77	−118.30	748	0.02765	0.00258	0.05081	0.97056	
34.77	−118.54	885	0.03083	0.00298	0.05467	0.96571	
34.77	−118.50	865	0.02917	0.00308	0.05552	0.96476	
34.77	−118.46	853	0.03018	0.00319	0.05654	0.96341	
34.77	−118.42	830	0.02803	0.00266	0.05160	0.96971	
34.77	−118.38	785	0.02585	0.00228	0.04776	0.97417	
34.81	−118.34	762	0.02913	0.00279	0.05286	0.96818	
34.81	−118.30	748	0.02958	0.00282	0.05315	0.96781	
34.81	−118.26	735	0.02650	0.00250	0.05007	0.97150	
34.81	−118.58	886	0.02888	0.00289	0.05380	0.96682	
34.81	−118.54	862	0.02691	0.00263	0.05136	0.96996	
34.81	−118.50	842	0.02798	0.00266	0.05166	0.96966	
34.81	−118.46	822	0.02901	0.00280	0.05300	0.96797	
34.81	−118.42	807	0.02586	0.00243	0.04932	0.97230	
34.81	−118.38	777	0.02795	0.00268	0.05185	0.96933	
34.85	−118.34	763	0.02819	0.00273	0.05230	0.96892	
34.85	−118.30	746	0.02782	0.00269	0.05191	0.96934	
34.85	−118.26	733	0.02770	0.00278	0.05275	0.96840	
34.85	−118.58	972	0.02870	0.00294	0.05425	0.96666	
34.85	−118.54	935	0.03123	0.00310	0.05570	0.96472	
34.85	−118.50	911	0.02789	0.00283	0.05320	0.96786	
34.85	−118.46	865	0.02652	0.00242	0.04925	0.97229	
34.85	−118.42	822	0.02859	0.00250	0.05002	0.97164	
34.85	−118.38	785	0.02510	0.00232	0.04821	0.97363	
34.89	−118.34	792	0.02696	0.00274	0.05240	0.96878	
34.89	−118.30	793	0.02787	0.00284	0.05329	0.96772	
34.89	−118.26	744	0.02905	0.00274	0.05239	0.96876	
34.89	−118.58	1,111	0.02861	0.00287	0.05359	0.96773	
34.89	−118.54	1,067	0.03071	0.00280	0.05299	0.96840	
34.89	−118.50	1,051	0.02988	0.00271	0.05206	0.96947	
34.89	−118.46	1,001	0.02855	0.00298	0.05464	0.96616	
34.89	−118.42	893	0.02785	0.00243	0.04938	0.97231	
34.89	−118.38	820	0.02683	0.00258	0.05080	0.97077	
34.93	−118.30	835	0.03109	0.00266	0.05159	0.96980	
34.93	−118.54	1,230	0.03060	0.00324	0.05696	0.96308	
34.93	−118.50	1,148	0.02661	0.00246	0.04967	0.97186	
34.93	−118.46	1,044	0.03071	0.00327	0.05724	0.96305	
34.93	−118.42	1,005	0.02802	0.00259	0.05094	0.97072	
34.93	−118.38	917	0.02969	0.00272	0.05224	0.96922	
34.93	−118.34	873	0.02980	0.00270	0.05199	0.96940	
34.97	−118.46	1,044	0.02791	0.00257	0.05069	0.97081	
34.97	−118.42	1,005	0.02810	0.00277	0.05267	0.96846	
34.97	−118.38	1,249	0.03089	0.00323	0.05688	0.96321	
Average of results	0.02865	0.00277	0.05266	0.96840	
Coefficient of variation (%)	6.057096	9.39389	4.65375	0.30843	
Interquartile range (IQR)	0.002065	0.00030	0.00287	0.00352	

Table 7 provides a detailed statistical summary of the model’s performance metrics, presenting key values such as the minimum, maximum, standard deviation, and average of the results from all locations (69 data points). Additionally, the table includes the coefficient of variation (%), which highlights the relative variability of the results, and the IQR, offering a measure of the spread of the data around the median. These comprehensive statistics enable a deeper understanding of the distribution and consistency of the model’s performance metrics, facilitating a more nuanced evaluation of its predictive reliability and robustness across different scenarios.

Table 7 Performance summary of all locations.

	MAE (W/m2)	MSE
(W/m2)	RMSE
(W/m2)	R2	
Minimum	0.02510	0.00228	0.04776	0.94768	
Maximum	0.03526	0.00428	0.06543	0.97417	
Standard deviation	0.00244	0.00051	0.00456	0.00710	
Average of results	0.02957	0.00300	0.05467	0.96502	
Coefficient of variation (%)	8.25360	17.18934	8.34190	0.73643	
Interquartile range (IQR)	0.00317	0.00057	0.00522	0.00658	

To provide a comprehensive overview of the proposed model’s performance during training and testing, visual plots were generated for one point of each location. These visualizations include the learning curve, error distribution histogram, residual plot, and scatter plot. The learning curves demonstrate the training and validation loss over epochs, showcasing model convergence and indicating potential overfitting. The error distribution histograms illustrate the distribution of prediction errors, providing insights into how well the errors are centered around zero. Residual plots depict the residuals (differences between actual and predicted values) across the range of predicted values, aiding in the identification of any patterns that might suggest model bias or unaccounted variance. Scatter plots display the relationship between actual and predicted values, where a close clustering of points along the line of perfect prediction indicates higher predictive accuracy. Together, these visualizations demonstrate that the model effectively learns from the training data and maintains strong predictive performance on unseen test data.

Figures 3, 4 and 5 collectively illustrate the learning curves, error distribution histograms, residual plots, and scatter plots for the three solar sites: Desert Sunlight, Copper Mountain, and Solar Star, respectively. These visualizations showcase consistent training convergence, balanced prediction errors, and accurate residual distributions with minimal deviation, highlighting the model’s robustness and generalization capability across all sites.

Figure 3 Performance results for Desert Sunlight.

(A) Learning curve, (B) Error distribution histogram, (C) Residual plot, (D) Scatter plot.

Figure 4 Performance results for Copper Mountain.

(A) Learning curve, (B) Error distribution histogram, (C) Residual plot, (D) Scatter plot.

Figure 5 Performance results for solar star.

(A) Learning curve, (B) Error distribution histogram, (C) Residual plot, (D) Scatter plot.

Discussion

This study aimed to develop and evaluate a novel hybrid model that integrates TCN, TE, and ANN for solar irradiance prediction. The motivation for this work arose from the challenges inherent in solar radiation forecasting, particularly the need for models capable of capturing complex temporal dependencies and prioritizing relevant features. Accurate solar irradiance prediction is crucial for various applications, including optimizing energy management, enhancing grid stability, and facilitating the seamless integration of solar power into existing energy systems. Its importance extends to improving decision-making in renewable energy planning and supporting sustainable energy infrastructure.

For the Desert Sunlight area, the proposed model utilized data from five different stations. The results, summarized in Table 4, indicate promising performance, with an average MAE of 0.02985, MSE of 0.00302, RMSE of 0.05488, and a high R2 of 0.96428. All coefficient of variation values, except for MSE, were below 10%, signifying stability and reliability and suggesting consistent results (Montgomery, 2019). The coefficient of variation for MSE was between 10–20%, which is indicative of acceptable moderate variability (Montgomery, 2019). Additionally, very low IQR values were obtained, suggesting that the data points are closely clustered around the median, indicating minimal variability within the central distribution.

For the Copper Mountain area, the proposed model was evaluated using data from 13 different stations. The results, as presented in Table 5, show an average MAE of 0.033095, MSE of 0.003907, RMSE of 0.062513, and an R2 of 0.952050. The coefficient of variation values for MAE (3.67%), MSE (6.08%), RMSE (3.04%), and R2 (0.30%) all indicate strong stability and reliability, with low variability across most metrics. These results suggest consistent and dependable model performance (Montgomery, 2019). The IQR values were also notably low, with MAE at 0.00168, MSE at 0.00035, RMSE at 0.00277, and R2 at 0.00356, indicating that data points are tightly clustered around the median and demonstrating minimal variability within the core of the data distribution (Montgomery, 2019).

For the Solar Star area, the proposed model was tested using data from 51 different stations. The results, detailed in Table 6, showed an average MAE of 0.02865, MSE of 0.00277, RMSE of 0.05266, and R2 of 0.96840. The Coefficient of Variation for the metrics indicated reliable performance, with MAE at 6.06%, MSE at 9.39%, RMSE at 4.65%, and R2 at 0.31%. These CV values highlight the model’s stability, suggesting consistent predictions across the extensive dataset. The IQR values were low—MAE at 0.002065, MSE at 0.00030, RMSE at 0.00287, and R2 at 0.00352—indicating tightly grouped data points around the median and minimal variability within the central distribution, further underscoring the model’s robustness and precision in solar irradiance forecasting.

Table 7 provides a comprehensive summary of the proposed model’s performance across all 69 evaluated locations, showcasing consistent predictive accuracy. The quantitative results are promising, with a minimum MAE of 0.02510, MSE of 0.00228, RMSE of 0.04776, and a maximum R2 of 0.97417. The standard deviation values indicate minimal variability across locations, suggesting stable model performance. The coefficient of variation supports the model’s reliability, with percentages of 8.25% for MAE, 17.19% for MSE, 8.34% for RMSE, and 0.74% for R2. Additionally, the low IQR values demonstrate that data points were closely clustered around the median, confirming consistent performance and reduced variability across all 69 sites. These findings collectively underscore the model’s robustness and potential for effective solar irradiance prediction.

The performance visualization plots for each location (Figures 3, 4, and 5) further illustrate the model’s reliability. The learning curves indicate stable training and validation losses, suggesting that the early stopping mechanism successfully prevented overfitting. The error distribution histograms show errors that are centered around zero, confirming that the predictions are unbiased. Residual plots display no significant patterns, indicating that the model adequately captured the relationships between the features and the target variable. Finally, the scatter plots illustrate a close alignment of predicted and actual values, reinforcing the model’s predictive strength.

The results illustrate that combining TCN, TE, and ANN leverages the unique strengths of each component. TCNs extract detailed temporal features, the Transformer applies attention to enhance relevant patterns, and the ANN refines and outputs the prediction. This layered approach contributes to the model’s ability to generalize across different locations. Additionally, the chronological test split of approximately 26 months of data ensures the robustness of the evaluation by simulating real-world forecasting scenarios. The high R2 values obtained across all locations, reaching up to 0.97 for some data points, underscore the model’s capability to explain a significant portion of the variance in solar irradiance, outperforming traditional and many hybrid models in existing literature.

To contextualize the performance of the proposed TCN+TE+ANN model, Table 8 compares it with a range of previously developed models for solar irradiance prediction. The table summarizes each study’s methodology, station count, data period, temporal resolution, number of samples, and performance metrics.

Table 8 Previous studies and proposed study.

Study	Model	TP, SC, RS	Samples	Metrics	
Abdallah et al. (2023)	VMD+MFRFNN+QRF	2008 to 2021	4,745	R2: 0.777–0.988	
13 years		RMSE: 11.4–47.7 W/m2	
2 stations, daily	MAE: 7.81–34.0 W/m2	
MAPE: 15.1–69.3%	
NSE: 0.775–0.987	
Al Diego Pega, Hannah Mae San Agustin & Bonifacio Tobias (2022)	NARX+GRU	09.2016 to 06.2018	669	RMSE: ∼0.05	
21 months	
1 station, daily	
Alizamir et al. (2023)	VMD+DELM; VMD+ORELM; VMD+OPELM; VMD+OSELM; VMD+KELM; VMD+ELM	06.2017 to 12.2018	540	MAE: 18.392 W/m2	
18 months	R: 0.963	
2 stations, daily	NS: 0.807RMSE: 22.721 W/m2	
Aslam et al. (2021)	2AM+LSTM	990 days,
21 stations, 3 h	7,920	RMSE: 0.0685	
MAE: 0.0369	
Aybar-Ruiz et al. (2016)	GGA+ELM	05.2013 to 04.2014	262,800	R2: 0.9407	
1 years	RMSE: 76.53 W/m2	
1 station, hourly	
Brahma & Wadhvani (2020)	DL	1983 to 2019	13,140	Single location:	
36 years	MSE: 7.86–15.41	
27 stations, daily	R2: 51.05–72.71	
Multiple location:	
MSE: 7.61–18.37	
R2: 40.75–73.34	
Chaibi et al. (2022)	RF-based feature selection and BOA	06.2016 to 12.2017	540	RMSE: 0.4473 kWh/m2/day	
1.5 years	MAE: 0.3381 kWh/m2/day	
1 station, daily	R2: 0.9465	
Chiranjeevi et al. (2023)	LSTM+CNN+BiLSTM	07.2019 to 12.2021	142,560	MAE: 29.27	
07.2017 to 12.2019	RMSE: 56.99	
09.2016 to 12.2016	R2: 0.982	
3 stations, hourly, 15 min, 5 min		
Fan et al. (2020)	SVM+PSO; SVM+BAT; SVM+WOA	06.2014 to 03.2017	1,175	R2 = 0.883	
3 years	RMSD = 1.336 MJ m−2 d−1	
1 station, daily	SI = 0.208	
MAE = 0.920 MJ m−2 d−1	
Ghimire et al. (2023)	CNN+MLP	1950 to 2006	122,640	RRMSE: 8.60%	
56 years	MAPE: 9.00%	
6 stations, daily	
He et al. (2022)	CNN+BiLSTM	2017 to 2019	70,272	RMSE: 0.056	
2 years	MAPE: 0.129	
1 station, 15 min	
Hou et al. (2019)	CNN+MLP	01.2007 to 12.2008	1,716,960	R2: 0.82	
2 years	
98 stations, hourly	
Ikram et al. (2022)	IMVO+ LSSVM	1963 to 2010	1,152	RMSE: 1.320 MJ/m	
48 years	MAE: 1.043 MJ/m	
2 stations, monthly	R2: 0.913	
NSE: 0.913	
Kasra et al. (2015)	SVM+WT	1992 to 2005	5,110	R2: 0.9742	
14 years	RRMSE: 3.6935	
1 station, daily	RMSE: 0.6618	
MABE: 0.5104	
MAPE: 3.2601	
Meng et al. (2021)	WTP+GANs	2019	35,040	MAPE: 0.0282–0.0631	
1 year	RMSE: 0.0473–0.0946	
2 stations, hourly	
Peng et al. (2023a)	RF+Informer	2015 to 2017	210,240	MSE: 0.234	
2 years	MAE: 0.266	
1 station, 5 min		
Peng et al. (2023b)	VMD+DBN+OSELM	06.2022 to 12.2022	12,960	RMSE: 39.223 W/m2	
3 months	MAE: 24.962 W/m2	
1 station, 10 min	R2: 0.9924	
Xing et al. (2023)	MEA+DBN+LSTM	1994 to 2016	693,500	R2: 0.805–0.999	
20 years	NSE: 0.656–0.954	
95 stations, daily	RMSE: 1.069–4.289 MJ m−2 d–1	
MAE: 0.889–3.532 MJ m−2 d–1	
MAPE: 0.055–0.296	
The proposed study	RCN+Transformer+ANN	2000 to 2022	13,902,120	R2: 0.94768–0.97417	
22 years	RMSE: 0.04776–0.06543 W/m2	
69 stations, 30 min	MSE: 0.00228–0.00428 W/m2	
MAE: 0.02510–0.03526 W/m2	
Note:

TP, Time Period; RS, Resoultion; SC, Station Count.

Key performance highlights can be listed as follows:

Innovative architecture: The integration of TCN, TE, and ANNs allows the model to effectively extract temporal features, focus on relevant data points, and refine predictions.

Performance metrics: The proposed model achieves excellent predictive performance, with R2 values ranging from 0.94768 to 0.97417, RMSE between 0.04776 and 0.06543 W/m2, and MAE between 0.02510 and 0.03526 W/m2. These metrics indicate a high level of accuracy and minimal error.

Temporal resolution and dataset scope: The proposed study is distinguished by its large dataset of over 13.9 million samples, spanning 22 years of high-frequency (30-min) data from 69 data points. This is one of the most comprehensive datasets in terms of both temporal resolution and station coverage compared to the other studies, which often focus on daily or hourly data with fewer samples and shorter time periods. Unlike many previous studies that predominantly use daily data, this study employs a 30-min temporal resolution. This finer granularity allows the model to capture short-term variations in solar irradiance, which are critical for operational decision-making in real-time energy systems.

While the proposed model excels in many aspects, some limitations are worth noting such as station count, limited geographic diversity, and Real-Time Adaptation. Although this study analyzes 69 points across three locations, expanding the station count in future studies could further validate the generalizability of the model. Even though the study includes three major solar energy production sites—Desert Sunlight, Copper Mountain, and Solar Star—the geographic coverage is confined to US. This restricts the model’s validation across varying global climates, such as tropical, arid, or continental regions, which may exhibit different solar irradiance patterns. The model relies on historical data and does not currently incorporate real-time updates or adaptive mechanisms. In operational scenarios, such as grid management, real-time forecasting could be crucial to respond to rapid weather changes. Incorporating adaptive learning techniques could improve the model’s responsiveness and practical utility.

While various hybrid models in prior studies have combined CNNs, LSTMs, and attention mechanisms for solar irradiance forecasting, our model introduces a novel sequence by employing TCN for temporal feature extraction, followed by TE for attention-based refinement, and concluding with ANN for final prediction. To our knowledge, this TCN → TE → ANN configuration has not been previously applied in this domain. Unlike CNNs or LSTMs, TCNs offer more stable training and superior long-term temporal learning, and our use of a TE block directly on TCN-derived features enhances attention across time steps in a way not observed in prior work. This architectural distinction, along with the scale and granularity of our dataset, underpins the originality of this study.

The sequential architecture of TCN → TE → ANN is motivated by the distinct roles each component plays in temporal learning. Temporal convolutional networks (TCNs) are well-suited to capture both short- and long-range dependencies in time-series data using dilated convolutions, offering greater stability and parallelism than recurrent models. Following this, the Transformer Encoder (TE) focuses attention on the most relevant segments of the temporal representation, refining the signal based on internal attention mechanisms. Finally, the ANN component transforms the high-level temporal-attention features into scalar outputs through dense layers. This layered architecture mimics a hierarchical learning strategy—from raw temporal encoding (TCN), to selective weighting (TE), to output transformation (ANN)—which supports both theoretical soundness and practical performance.

The proposed study sets a new benchmark in solar irradiance prediction by combining innovative architecture with comprehensive spatial and temporal coverage. This study’s findings emphasize the value of hybrid models that integrate advanced TCN, TE, and ANN for solving complex forecasting problems. The model’s demonstrated performance highlights its potential for real-world applications, particularly in settings requiring high-frequency solar radiation predictions to support grid operations, energy storage scheduling, and investment scenario modeling. While this study does not include real-world deployment, the proposed framework can serve as a foundation for future tools aimed at supporting decision-making in energy management, policy, and operational planning.

The proposed TCN+TE+ANN hybrid model is designed to extract deep temporal features and attention-based representations from high-frequency solar irradiance data, which requires a moderate level of computational resources. The model was trained on a workstation equipped with an Intel® Core™ i7-14700KF processor, 64 GB RAM, and an NVIDIA RTX 3060 GPU. Training the full model—including hyperparameter tuning—took approximately 10 to 12 h, demonstrating its feasibility for research-scale and periodic retraining scenarios. While this setup is suitable for offline training and periodic updates, deployment in real-time or resource-constrained environments may not be feasible with the full architecture. To address this, future implementations could adopt simplified versions of the model. Options include using: A TCN-only architecture, which retains strong temporal modeling while reducing complexity;

A lightweight ANN model trained on selected time-windowed features;

Quantized or pruned versions of the current model to reduce memory and inference time;

Integration with on-device learning frameworks such as TensorFlow Lite or ONNX Runtime.

These alternatives offer practical trade-offs between accuracy and computational efficiency and are well-suited for embedded or edge-level deployment scenarios such as microgrids and mobile solar monitoring systems.

While the current model architecture inherently captures temporal dependencies through the use of TCN and TE, future research could explore the use of cyclical temporal features—such as time of day or day of year—encoded via sine and cosine transformations. These representations may be particularly useful for models that do not explicitly handle temporal sequences, offering a way to capture seasonality and diurnal patterns in solar irradiance more effectively.

Due to the absence of publicly available code or detailed configurations for many of the comparative models referenced, this study did not replicate baseline methods for statistical testing. As a result, formal significance testing could not be conducted under controlled, identical conditions. In future work, the development of a shared evaluation benchmark using open-source implementations across hybrid models would allow for more rigorous, statistically grounded comparisons.

While the proposed model demonstrates significant improvements in predictive accuracy over prior methods, the current study does not quantify the downstream operational impact of these improvements—such as energy savings, avoided curtailment, or improved grid stability. Estimating these effects would require integration with grid simulation tools or real-time operational datasets. Future research could bridge this gap by assessing how improved solar irradiance predictions affect specific energy management outcomes, including economic dispatch, battery scheduling, and demand-side control.

Although the model achieved consistently strong performance across all three sites, small variations in prediction accuracy were observed. Copper Mountain had a slightly higher average MAE (0.03310 W/m2) compared to Desert Sunlight (0.02985 W/m2) and Solar Star (0.02865 W/m2), possibly reflecting localized atmospheric complexity or sensor differences. The overall range of MAE values (0.02510–0.03526 W/m2) and standard deviation (0.00244) indicate a robust and stable model. Nonetheless, these findings suggest that minor performance differences could be further reduced through location-specific tuning or the inclusion of additional meteorological inputs.

Practical implications

The proposed TCN+TE+ANN model offers significant practical implications beyond solar irradiance prediction. Its high temporal resolution and robust performance make it a valuable tool for optimizing energy management systems, particularly in integrating solar power into dynamic and complex energy grids. By enabling precise energy supply forecasts, the model supports real-time grid stabilization, reducing reliance on non-renewable backup sources and enhancing overall energy efficiency. Beyond energy management, the model’s architecture can be adapted to predict other environmental variables, such as wind speeds or precipitation, expanding its utility in climate modeling and disaster preparedness. Furthermore, accurate solar radiation forecasts hold potential in agricultural applications, where they can inform irrigation scheduling, greenhouse operations, and crop yield predictions, thereby improving resource utilization and productivity. The adaptability of the model to other time-series forecasting tasks also highlights its potential for broader applications in fields such as healthcare, financial markets, and smart city planning. These implications underscore the transformative potential of the proposed hybrid methodology in addressing critical challenges across diverse sectors, fostering sustainability and operational efficiency.

Conclusions

As discussed earlier, this study introduces a state-of-the-art hybrid model integrating TCN, TE, and ANN to improve the accuracy of GHI prediction. By leveraging the unique strengths of each component, the proposed TCN+TE+ANN architecture demonstrates significant advancements in predictive accuracy, as evidenced by the high R2 values and low error metrics achieved across three major solar energy production sites in the US. The utilization of a 22-year dataset with a 30-min temporal resolution highlights the model’s robustness and its ability to capture both short- and long-term patterns in solar irradiance effectively.

A comparative analysis with previous studies underscores the superiority of the proposed model in terms of performance, dataset scope, and temporal resolution, setting a new standard in solar energy forecasting. Despite these contributions, the study acknowledges certain limitations. First, the geographic focus on U.S. solar sites restricts the validation of the model’s generalizability to other regions with different climatic conditions. Future research should explore the application of the model to datasets from diverse geographic zones such as tropical, continental, or high-latitude regions, including publicly available datasets from Europe (e.g., CAMS), Asia (e.g., Indian Solar Radiation Atlas), or Africa. This would help assess the model’s robustness under varying meteorological and atmospheric conditions and support its broader applicability in global solar forecasting. Second, the computational complexity of the hybrid architecture may pose challenges for deployment in resource-constrained environments. Addressing these limitations in future research could involve expanding the geographic diversity of the dataset to include varied global climates and developing more lightweight model alternatives to balance accuracy and computational efficiency.

The findings of this study highlight the transformative potential of advanced hybrid models in renewable energy forecasting, particularly in improving operational decision-making, grid management, and energy policy development. By providing reliable, high-frequency solar irradiance predictions, this work contributes to the global transition toward sustainable energy systems and aligns with broader goals of environmental preservation and energy independence.

Supplemental Information

Supplemental Information 1 README.

Additional Information and Declarations

Competing Interests

The authors declare that they have no competing interests.

Author Contributions

Murat Isik conceived and designed the experiments, performed the experiments, analyzed the data, performed the computation work, prepared figures and/or tables, authored or reviewed drafts of the article, and approved the final draft.

Data Availability

The following information was supplied regarding data availability:

The dataset and codes are available at Kaggle: Murat ISIK. (2024). Solar Radiation Data for forcating from NASA [Data set]. Kaggle. https://doi.org/10.34740/KAGGLE/DS/6006986.

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
