# Peer review of "A novel hybrid TCN-TE-ANN model for high-precision solar irradiance prediction"

_PeerJ Computer Science, doi:10.7717/peerj-cs.3026_

## Round 0.1 · original submission · Major Revisions

Dear authors,

You are advised to critically respond to all comments point by point when preparing an updated version of the manuscript and while preparing for the rebuttal letter. Please address all comments/suggestions provided by reviewers, considering that these should be added to the new version of the manuscript.

Kind regards,
PCoelho

·

Basic reporting

The manuscript is well-structured and follows a logical progression from introduction through methodology to results and conclusions. The author provides comprehensive context regarding the importance of solar irradiance prediction for renewable energy applications, situating the research within the broader field effectively.

The introduction clearly articulates the research motivation and objectives. The literature review is extensive and up-to-date, providing a thorough overview of existing approaches to solar irradiance prediction, with particular attention to hybrid models combining various machine learning techniques.

The paper is generally well-written, though there are occasional minor grammatical issues that do not significantly impede understanding.

Experimental design

The author uses a comprehensive dataset spanning 22 years with 30-minute temporal resolution across 69 data points at three major US solar energy sites, providing substantial data for training and validation. The model architecture is well-conceived, combining three advanced techniques (TCN, TE, ANN) with clear technical justification for each component. The evaluation uses multiple appropriate metrics (MAE, MSE, RMSE, R²) to assess performance comprehensively.

However, there are notable deficiencies in the manuscript.
-No statistical significance testing: The paper lacks statistical tests to determine if performance improvements over baseline or comparative models are statistically significant.
-Despite mentioning hyperparameter tuning, the final hyperparameter values chosen for each component are not clearly tabulated.
- There's no discussion of whether features were normalized, standardized, or otherwise scaled, which is critical for neural network training.
- While the paper mentions that hyperparameters were selected based on "lowest validation error," the specific decision process between competing model configurations is not clearly explained.

Validity of the findings

The paper claims novelty through the specific combination of Temporal Convolutional Networks (TCN), Transformer Encoders (TE), and Artificial Neural Networks (ANN) for solar irradiance prediction.

However, the novelty assessment has several limitations;
- While the TCN-TE-ANN integration appears novel, the paper doesn't sufficiently distinguish this from existing hybrid approaches. Many cited works already combine convolutional approaches with attention mechanisms and neural networks in various configurations.
-The paper focuses on empirical performance without developing new theoretical insights about why this specific combination works better than others. The rationale for the specific architectural choices remains primarily practical rather than theoretical.
- There's no clear statement of how much better the proposed approach is compared to simpler methods in terms of practical impact (e.g., reduced energy wastage percentages).

The conclusions in the paper are generally well-stated and appropriately limited to the supporting results. However;
- The conclusion states that the model contributes to "informed decision-making in energy management, policy, and investment" without demonstrating actual implementation in these contexts.
- The paper doesn't analyze when and where the model performs poorly.
- The paper promises to contribute to "informed decision-making in energy management, policy, and investment," but provides no actual implementation examples or case studies.

In conclusion, while the paper presents interesting results, it has significant shortcomings in establishing novelty, demonstrating practical impact, providing a comprehensive evaluation, and outlining a clear path for future research.

Reviewer 2 ·

Basic reporting

The related work section is thorough, citing a wide array of hybrid models and deep learning techniques relevant to the domain.

Experimental design

Justify the removal of datetime features (year, month, hour, etc.), as these may improve seasonality modeling.

Validity of the findings

Discuss the computational requirements of the final model and suggest lightweight alternatives for real-time applications.

Summarize performance comparisons more clearly in a tabular form with consistent metrics.

Additional comments

Fix encoding issues (e.g., “TE9s”) and small grammatical errors.

Future work should test the model on solar datasets from regions beyond the U.S.

Cite this review as

---

## Round 0.2 · accepted · Accept

Dear authors, we are pleased to verify that you meet the reviewer's valuable feedback to improve your research.

Thank you for considering PeerJ Computer Science and submitting your work.

Kind regards
PCoelho

·

Basic reporting

The reporting has been enhanced to comply with the requirements of the paper.

Experimental design

Almost all applicable change requests in the experimental design were reflected in the revised version of the manuscript. Some of the demands were rejected by the author, due to inability to compare the studies statistically. Although, inability reduces the quality of the paper in presenting, it seems feasible especially in this area as pointed by the author.

All necessary, improvements were performed now.

Validity of the findings

The article is good enough to contribute to the literature, with updated explanations, and presented data.